# Numerical Optimization of Electromagnetic Performance and Aerodynamic Performance for Subsonic S-Duct Intake

**Bin Wang and Qiang Wang \***

School of Energy and Power Engineering, Beihang University, Beijing 100191, China
* Correspondence: qwang518@buaa.edu.cn

**Abstract:** In order to improve the performance of subsonic unmanned aerial vehicle (UAV), a knapsack S-duct intake has been designed. The influences of an S-bend diffuser on aerodynamic performance and electromagnetic performance were analyzed firstly. The viscous flow field has been simulated by solving Favre averaged Navier–Stokes equations using a shear stress transport (SST) k-ω turbulence model. The surface current has been simulated by solving Maxwell equations using a multi-level fast multipole method (MLFMM). The multi-objective optimization of the S-duct intake was studied by using the diffuser as the optimized object. The parametric expression of the diffuser model was realized using the fourth order function geometric representation technique. The efficient model based on the Kriging model and non-dominated sorting genetic algorithm-II (NSGA-II) were used to accelerate the optimization progress. By analyzing the results of an optimal intake chosen from the Pareto front, the total pressure distortion (TPD) index DC60 has decreased by 0.24 at the designed Mach number of 0.9, and the average Radar Cross Section (RCS) has decreased by 2db at the frequency of 3GHz. The optimized S-duct intake could have both excellent aerodynamic performance and electromagnetic performance at various complex conditions.

**Keywords:** S-duct intake; multiobjective optimization; efficient model; total pressure distortion; radar cross section; flow coefficient

## 1. Introduction

Recently, various unmanned aerial vehicles have been designed all over world with the rapid development of related technology. Considering that the intake is one of the main electromagnetic scattering centers of aircrafts [1], the design of intakes has become a technical difficulty in UAV research. The S-duct intake is usually designed such as a short-distance diffuser, which is prone to secondary flow separation and flow distortion, resulting in the loss of aerodynamic performance. Therefore, it is of great significance to carry out the aerodynamic and electromagnetic integrated optimization design for bump S-duct intakes.

In early stages, experiments were the major methods in the S-duct intake studies [2–5]. With the development of numerical calculation technology, CFD has been one of the most promising and practical methods for intake design [6]. Turbulence models used in S-duct intake simulations were thoroughly investigated, indicating that the SST k-ω turbulence model is suitable for precisely simulating secondary flow and flow separation [7,8]. After the optimization design concept is proposed, optimization algorithm coupling with CFD simulation becomes a typical research approach in intake design [9,10]. Knight et al. [9] established the automatic optimization design method of three-dimensional S-duct intake, improving the design efficiency and reducing the TPD of the outlet. HyoGil et al. [11] established the whole local optimization method of S-duct diffusers based on the Krigin agent model, indicating that the Krigin model has a satisfactory prediction accuracy for the aerodynamic performance of intake. Gan and Zhang [12] conducted a three-dimensional diffusing S-duct intake optimization by using the multi-objective optimization strategy, reducing

the computation cost, and causing the intake to avoid overexpansion. Zhang et al. [13] adopted the adjoint optimization method to optimize the design parameters of the S-duct intake, causing the curvature and section areas to reduce the TPD of the outlet.

In electromagnetics research, CEM methods have been developed to calculate RCS, which is commonly used as an evaluation index of electromagnetic scattering intensity [14]. Chung et al. [15] compared the RCS of an intake and an intake with a corner reflector inside by using MLFMM, proving that the engine components have a significant influence on the RCS of intake. Vogel [16] presented the RCS for a fighter with an engine intake including fan blades using MLFMM, indicating that the intake and blades could increase the echo intensity of the fighter. Shen et al. [17] compared the results of the ordinary intake and the S-duct intake by using MLFMM, proposing that the S-duct can effectively reduce the RCS of intake.

However, the two performances of the S-duct intake are generally studied independently, resulting in the lack of practical value for some conclusions. Therefore, to reduce both the TPD index at the outlet and the RCS of the intake radiated by the plane electromagnetic wave with a frequency of 3GHz, an aerodynamic and electromagnetic integrated optimization design of the S-duct intake with a flight altitude of 12km and a Mach number of 0.9 was carried out, based on reliable numerical simulation methods and multi-objective genetic algorithms.

## 2. Computational Methodology

The model of the S-duct intake is shown in Figure 1 and the dimensions of inlet and outlet are shown in Table 1. In order to improve electromagnetic performance, the section of the diffuser changes from the semi-elliptic inlet to the circular outlet through the S-duct diffuser. To simulate the fuselage boundary layer, a flat plate with the length of 2 m and thickness of 0.01 m was placed in front of the inlet. Besides, the maximum cross-sectional area of the bump is 10% of the inlet to exclude fuselage boundary layer.

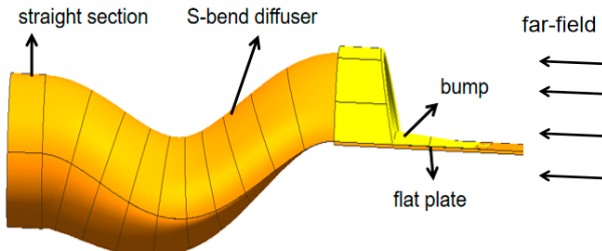

**Figure 1.** Model of S-duct intake.

**Table 1.** Dimensions of inlet and outlet.

| Variable | | Value |
|---|---|---|
| | Area $S_1/m^2$ | 0.55 |
| Inlet | High $H_1/m$ | 0.5 |
| | Sweep angle/° | 12.43 |
| Outlet | Area $S_2/m^2$ | 0.64 |
| | Diameter $D_2/m$ | 0.9 |
| Length of S-duct diffuser/m | | 2.5 |

### 2.1. Verification

For complex flow field simulation of the S-duct intake, the accurate turbulence model can effectively improve the computational accuracy of CFD. In order to verify the accuracy of the SST k-ω turbulence model mentioned above in calculating the transonic flow field, a typical intake structure was simulated numerically. Herrmann et al. [18] presented a

series of detailed experimental results for the intake internal flow field. According to the literature, an intake model with a throat length of 79.3 mm was used to validate the CFD method, facing a flow with a Mach number of 2.5, temperature of 295 K, pressure of 5.6 bar, and attack angle of 10°. The intake model and pressure measuring point positions are shown in Figure 2. The detailed parameters of the intake and the coordinates of five points marked in the figure are shown in Table 2.

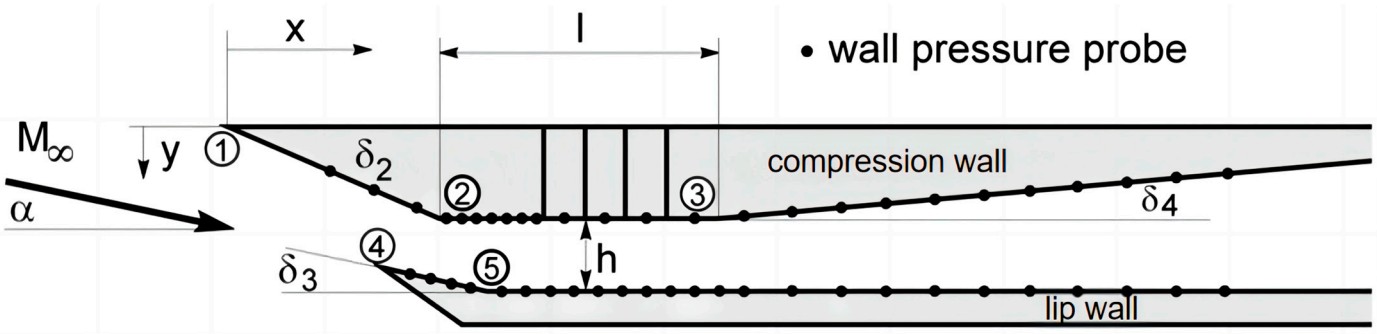

**Figure 2.** Intake model and pressure measuring point positions.

**Table 2.** Dimensions of the intake.

| Parameter | Point Coordinates | X/mm | Y/mm |
|---|---|---|---|
| Throat height h: 15 mm | 1 | 0 | 0 |
| Total length: 400 mm | 2 | 45.7 | 18.0 |
| Second compression angle $\delta_2$: 21.5 | 3 | 75.0–145.0 | 18.0 |
| Lip angle $\delta_3$: 9.5 | 4 | 35.0 | 29.0 |
| Expansion angle $\delta_4$: 5 | 5 | 58.9 | 33.0 |

The mesh near the wall were compressed to meet the requirements of the standard wall function. Figure 3 shows a comparison of the experiment and calculation results by the ratio of static pressure to total pressure on compression (a) and lip (b) walls. The calculation results are highly consistent with the experimental results. Therefore, the SST k-ω turbulence model was applied to a numerical simulation of the aerodynamic performance of the S-duct intake.

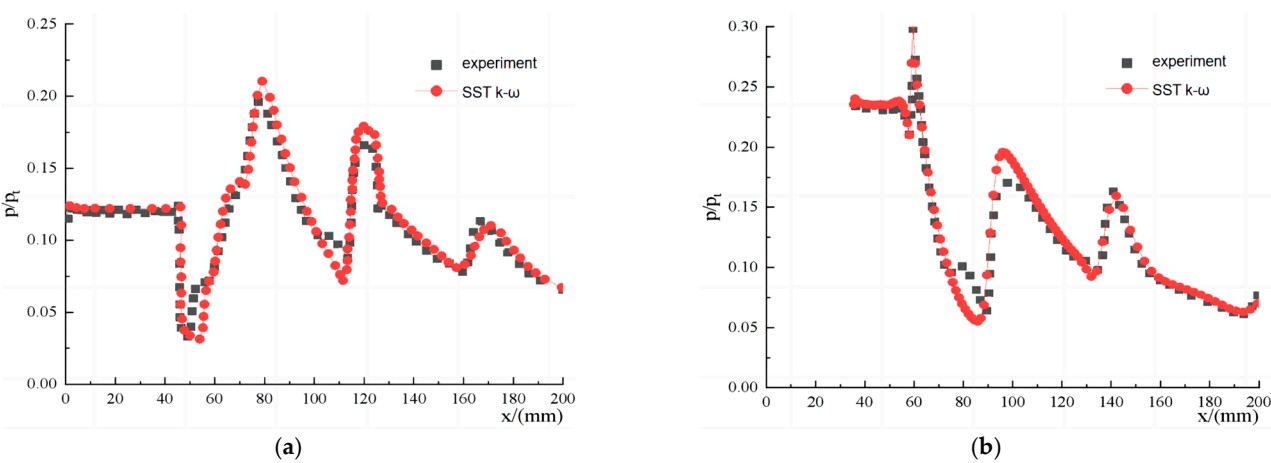

**Figure 3.** Wall pressure values of experiment and calculation. (**a**) Compression wall (**b**) Lip wall.

After constant research, various CEM methods for the RCS analysis of electrically large objects have been proposed [19]. As a numerical solution, MOM solves the Maxwell equations in integral form, resulting in a matrix too large to be used for RCS calculations

of electrically large objects. MLFMM reduces the memory required by turning the large matrix into several layers of smaller matrices, meaning that RCS calculations of electrically large objects re possible on a small workstation.

A variety of CEM methods were used to calculate a complex cylindrical cavity as shown in Figure 4 [20], including MOM, MLFMM, PO (physics optics method), and RL-GO (ray launching-geometrical optics method). The specific dimensions of the complex cylindrical cavity are shown in Table 3. The mesh size is 1/8 of wavelength.

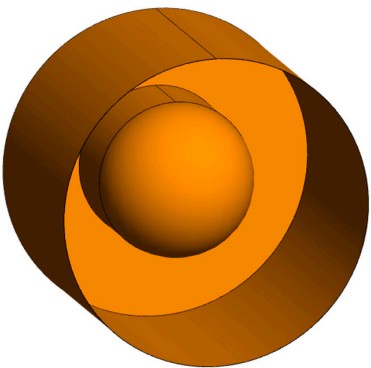

**Figure 4.** Complex cylindrical cavity.

**Table 3.** Dimensions of the complex cylindrical cavity.

| Variable | Value |
| --- | --- |
| Cavity diameter/m | 0.286 |
| Cavity length/m | 0.3 |
| Cylinder diameter/m | 0.16 |
| Cylinder length/m | 0.16 |

The frequency of incident wave was set as 5 GHz and the calculation angles were 0–50°, with intervals of 1°. The comparison between experimental results and calculation results of each CEM method is shown in Figure 5. Obviously, the MOM and the MLFMM have higher computational accuracy than the PO and the RL-GO with both polarization modes. It has been proved that the MLFMM requires less memory than the MOM. Therefore, the MLFMM was used to calculate the RCS of the S-duct intake.

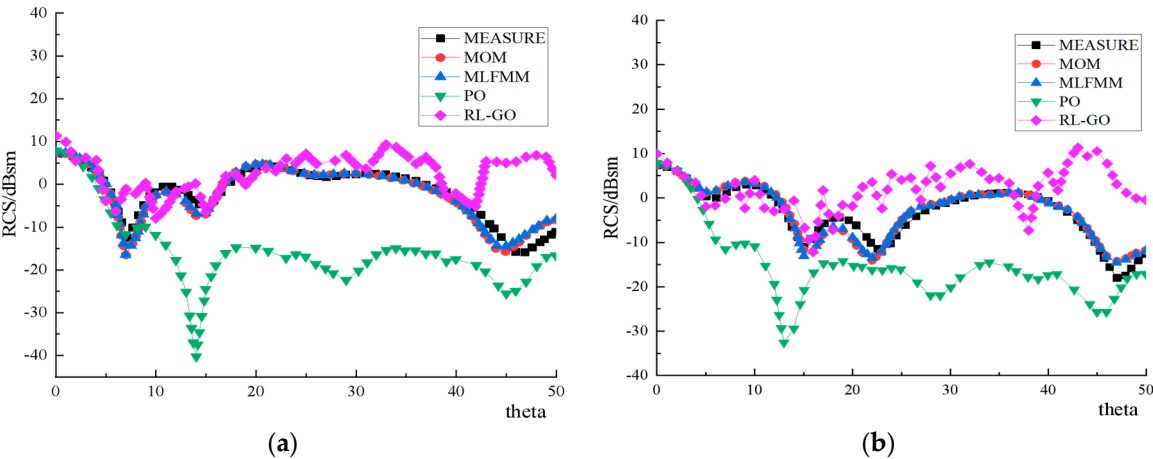

**Figure 5.** Results of RCS calculation and experiment. (**a**) Horizontal polarization (**b**) Vertical polarization.

### 2.2. Mesh Independence Study

The quality and quantity of mesh are crucial in fluid numerical simulation. In order to find out the most economic mesh number and ensure accuracy as well, three series of mesh were generated to validate the mesh independence, as shown in Table 4.

**Table 4.** Three types of mesh size.

| Mesh Type | Whole Mesh Number | Inner Mesh Number |
|---|---|---|
| Mesh-A | 1.04 million | 0.58 million |
| Mesh-B | 2.06 million | 1.23 million |
| Mesh-C | 4.03 million | 3.07 million |

The boundary condition of the inlet was defined as a pressure far-field, with a static pressure of 19,000 Pa, a Mach number of 0.9, and a static temperature of 218 K. The boundary condition of the outlet was defined as a pressure-outlet, with a static pressure of 25,000 Pa, and a total temperature of 253 K. The wall thickness of the three mesh were all 0.2 mm and the wall grid $y^+$ values ranged from 0 to 10.

The distribution of the Mach number at the symmetrical section calculated by using mesh-B is shown in Figure 6 and the total pressure distributions of the outlet calculated by using the three types of mesh are shown in Figure 7. Due to the large adverse pressure gradient in diffuser, two backflow zones appeared, reducing the actual flow area, and increasing the TPD index of the outlet. Besides, due to the large curvature of wall and the high subsonic mainstream, a local shock wave was formed at the upper wall of the S-bend section, reducing the total pressure recovery (TPR) coefficient. According to Figure 7, the difference of flow field distribution between mesh-A and mesh-B was more significant than that between mesh-B and mesh-C, especially at the boundary between the low- pressure and high-pressure zones below the outlet. The results of total pressure at outlet were summarized in Table 5. Therefore, the number of mesh used to calculate was set to 2 million such as mesh-B.

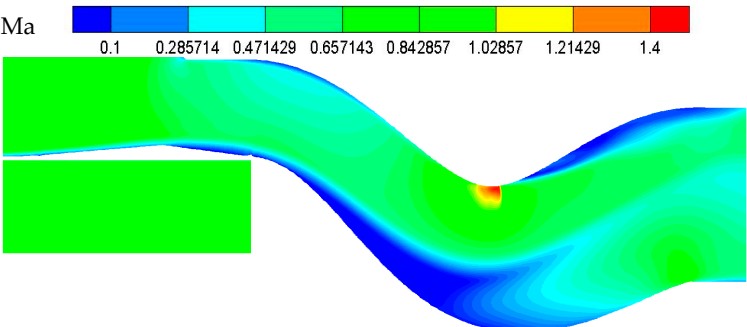

**Figure 6.** Mach number distribution of the symmetrical section.

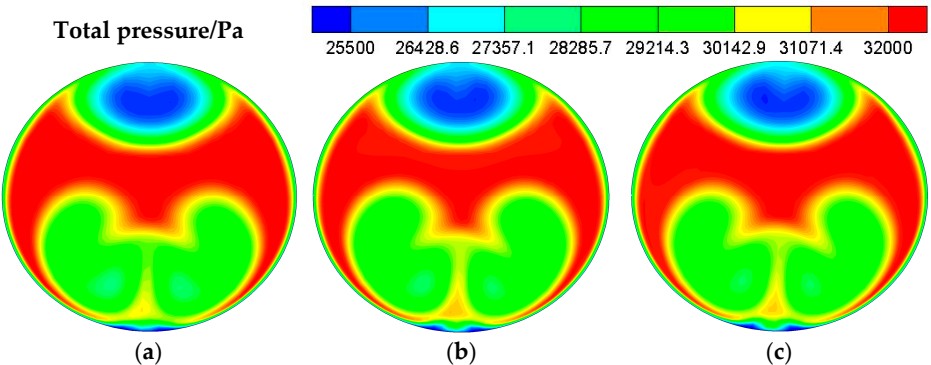

**Figure 7.** Total pressure distributions at outlet. (**a**) mesh-A (**b**) mesh-B (**c**) mesh-C.

**Table 5.** Total pressure statistics at outlet.

| Mesh Type | Total Pressure/Pa | | |
|---|---|---|---|
| | **Maximum** | **Minimum** | **Average** |
| Mesh-A | 32,249 | 25,123 | 30,113 |
| Mesh-B | 32,126 | 25,076 | 30,106 |
| Mesh-C | 32,150 | 25,068 | 30,120 |

### 2.3. Orientation Characteristics of RCS

Due to the piggyback layout of the S-duct intake, radar waves from the ground base station are generally obscured by fuselage. In order to test the electromagnetic scattering characteristics of the intake, the calculation angles were set as 0–30° in both vertical and horizontal directions, with intervals of 1°. The frequency of the plane wave was set as 3 GHz.

The intake model for RCS calculations is shown in Figure 8, with the plate located in front of the inlet removed. Two plates were placed on the lower and left side of the intake to simulate the shielding of electromagnetic waves from the airframe and the outlet was sealed off. The material was defined as an ideal electrical conductor. The mesh size was 1/8 of the wavelength. In order to universalize the results of the verification, three S-duct intakes of different parameters were calculated with both horizontal and vertical polarizations, respectively.

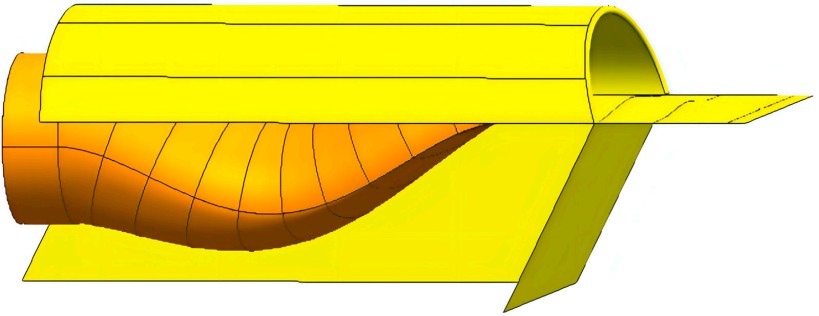

**Figure 8.** Intake model used for RCS calculations.

The RCS calculation results of model S are shown in Figure 9. Table 6 shows the average RCS values of the three models at four conditions. The conditions were abbreviated as a two-letter code, with the first letter indicating the polarization mode and the second indicating the direction of detection. With either polarization mode, the average RCS values are all greater than 11dB in a vertical direction and are less than 4dB in the horizontal direction.

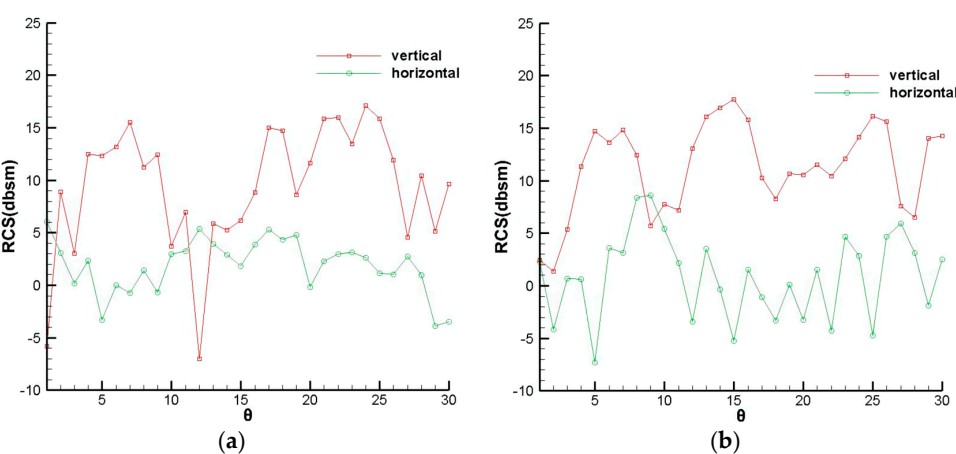

**Figure 9.** RCS calculation results of model S. (**a**) Horizontal polarization (**b**) Vertical polarization.

**Table 6.** Average RCS values of three models.

| Model | Average RCS/dbsm | | | |
|:---:|:---:|:---:|:---:|:---:|
| | **H-V** | **H-H** | **V-V** | **V-H** |
| S | 12.814 | 2.609 | 11.86 | 2.317 |
| K | 12.508 | 2.551 | 12.504 | 1.935 |
| P | 12.779 | 3.116 | 11.943 | 2.874 |

## 3. Method and Results of Optimization

### 3.1. Correlation Parameter

The first step of optimization design is to determine the parameters of the optimization scheme, including model parameters, objective functions, and constraints. There are three kinds of parameters to be defined to build an accurate S-duct intake model, including the center line function of the diffuser, the shapes of the diffuser sections, and the center deviation distance of the outlet. The shapes of the middle sections can be enclosed by two semi-elliptic curves of different short axis lengths, as shown in Figure 10. The area of any section can be calculated by Equation (1).

$$S_n = \frac{\pi}{2} b_n (a_{1n} + a_{2n}) \tag{1}$$

where n is a positive integer between 1 and 9, representing the nine middle sections. According to the equation, the shape of any middle section is determined by $S_n$, $a_{1n}$, and $a_{2n}$. The position and angle of any middle section vector are determined by the center line function of the diffuser.

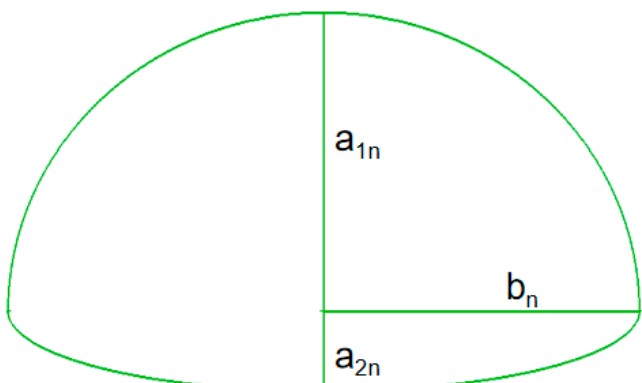

**Figure 10.** Shape of middle section.

Assume:

$$X_n = \frac{x_n}{x_{10}} \tag{2}$$

$$Y_n = \frac{y_n - y_0}{y_{10} - y_0} \tag{3}$$

where $x_n$ is an X-axis coordinate of the nth section center. $y_n$ could be a Y-coordinate of the nth section center or one of the three section shape parameters, including $S_n$, $a_{1n}$, and $a_{2n}$. To ensure the continuous variation of section parameters, there are four boundary conditions:

$$\text{if } X = 0, \ Y = 0 \text{ and } \dot{Y} = 0;$$

$$\text{if } X = 1, \ Y = 1 \text{ and } \dot{Y} = 0.$$

Therefore, the fourth order functions were used to express the variation of model parameters:

$$Y = AX^4 + BX^3 + CX^2 + DX + E \tag{4}$$

Substituting the boundary conditions into the equation:

$$Y = AX^4 - (2 + 2A)X^3 + (3 - A)X^2 \tag{5}$$

The four model parameter functions could be determined using four control variables. Therefore, a model of S-duct intake can be explicitly established by determining five variables, including the center line control variable A, the deviation of outlet center Y, the middle section area control variable $A_s$, the short axis length control variable of the upper semi-ellipse in middle section $A_1$, and the short axis length control variable of the lower semi-ellipse $A_2$.

In order to reduce the intensity of the local shock wave and effectively block the outlet, two constraints were determined during parametric modeling: ① the minimum Y-axis coordinate value of the connection between the upper wall vertices should be less than 0; ② the shortest distance between upper wall vertices of fifth and sixth sections should be greater than 50 mm. After testing, the two constraints mainly affect the independence of Y and A, as shown in Figure 11a. Therefore, a correction factor K is introduced to fix irrelevance between the parameters by replacing A, as shown in Figure 11b. The variation range of each parameter was determined according to design experience and quartic function characteristics, as shown in Table 7.

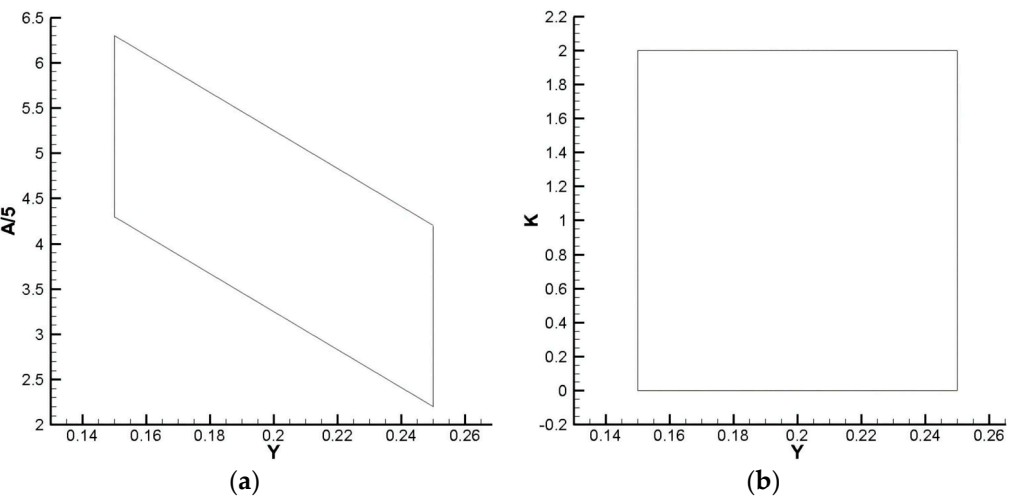

**Figure 11.** Value range of three parameters. (**a**) Y and A/5 (**b**) Y and K.

**Table 7.** Range of variations for each variable.

| Variable | Maximum Value | Minimum Value |
|:---:|:---:|:---:|
| K | 2 | 0 |
| Y | 0.25 | 0.15 |
| As | 3 | −3 |
| A1 | 3 | −3 |
| A2 | 3 | −3 |

### 3.2. Efficient Model and Error Analysis

In order to ensure the spatial distribution uniformity of the initial sample group, the Optimized Latin Hypercube method was used to select 75 points in sample space. The five performance parameters of each intake were calculated as shown in Table 8. $R_a$ is the

average RCS value in the vertical direction with horizontal polarization, while $R_b$ is the average RCS value in the vertical direction with vertical polarization. According to the table, there is a positive correlation between the TPR and the mass flow rate to some extent. The TPD has a greater impact on engine performance than the mass flow rate M. In order to improve the optimization efficiency, the maximum TPD index DC60 and the average RCS value R were taken as the main optimization objectives, with the constraint of M > 35 kg/s. DC60 could be calculated using Equation (6).

$$DC60 = \frac{\overline{P^*} - \overline{P^*_{min}(60)}}{\overline{q}} \tag{6}$$

where $\overline{P^*_{min}(60)}$ is the minimum average total pressure of the sector with an angle of 60° at the outlet. $\overline{q}$ is the average dynamic pressure at the outlet.

**Table 8.** Sample points and calculation results.

|  | Y | K | $A_s$ | $A_1$ | $A_2$ | DC60 | $R_a$ (dbsm) | $R_b$ (dbsm) | M (kg/s) | TPR |
|---|---|---|---|---|---|---|---|---|---|---|
| 1 | 0.1791 | 1.747 | 2.772 | 1.861 | 1.633 | 0.5676 | 12.81 | 11.86 | 31.392 | 0.9191 |
| 2 | 0.2411 | 0.506 | −1.861 | −1.177 | −2.241 | 0.2776 | 12.51 | 12.08 | 37.462 | 0.9470 |
| 3 | 0.212 | 1.241 | −1.785 | −3 | −1.481 | 0.4428 | 14.38 | 15.57 | 33.557 | 0.9287 |
| 4 | 0.1728 | 0.43 | −0.342 | 0.797 | −3 | 0.3590 | 12.80 | 15.14 | 37.748 | 0.9479 |
| 5 | 0.1652 | 0.506 | 0.494 | −2.392 | −2.392 | 0.2772 | 12.29 | 13.86 | 36.664 | 0.9384 |
| 6 | 0.1956 | 0.127 | −1.481 | −1.253 | 2.165 | 0.3114 | 11.40 | 15.15 | 37.567 | 0.9456 |
| 7 | 0.2361 | 0.329 | 2.089 | 1.633 | 1.861 | 0.2369 | 12.38 | 14.59 | 37.282 | 0.9430 |
| 70 | 0.1627 | 0 | −1.101 | −0.722 | −0.494 | 0.3747 | 13.26 | 13.48 | 37.067 | 0.9526 |
| 71 | 0.1513 | 1.342 | 1.937 | −2.089 | 1.177 | 0.5574 | 12.14 | 11.39 | 33.312 | 0.9273 |
| 72 | 0.1741 | 0.684 | −2.468 | −2.468 | 0.266 | 0.2934 | 14.39 | 12.48 | 36.736 | 0.9411 |
| 73 | 0.1614 | 1.418 | −1.709 | −1.937 | −2.013 | 0.4540 | 15.58 | 13.93 | 34.723 | 0.9310 |
| 75 | 0.1753 | 0.785 | 2.62 | 0.342 | −2.468 | 0.3106 | 13.51 | 14.07 | 37.363 | 0.9460 |

To improve optimization efficiency, four Kringing models between the performance variables and the model variables were established as the efficient models, based on initial sample data. The predicted values of the four efficient models and the actual values of the four performance parameters are shown in Figure 12. Obviously, only a few points have a large error, including five points in Figure 12a, two points in Figure 12b, and one point in Figure 12c. Therefore, the four effective models basically have high reliabilities in the sample space, which can effectively accelerate the optimization design.

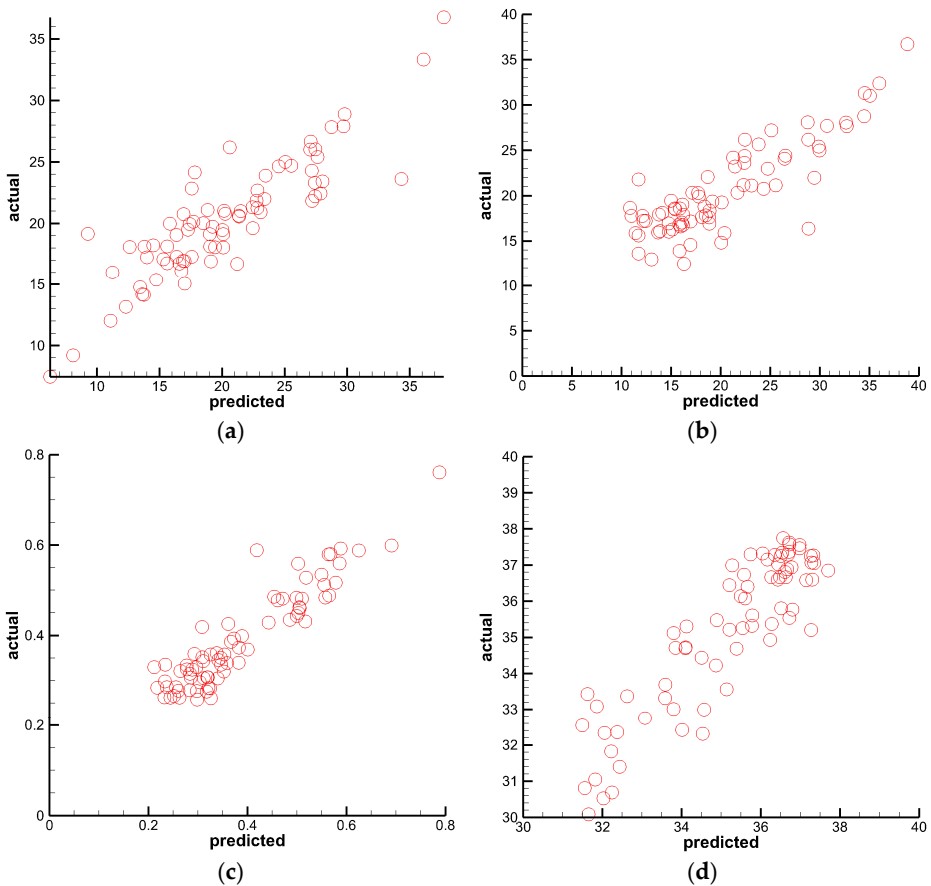

**Figure 12.** Scatter diagram of error analysis. (**a**) R$_a$ (**b**) R$_b$ (**c**) DC60 (**d**) M.

### 3.3. Pareto Front and Verification

The NSGA-II algorithm was used to optimize the diffuser by combining with the above effective models, with a population size of 200, a generations number of 20, a crossover probability of 0.9, and a mutation distribution index of 20. Figure 13 shows the gradual convergence of objective parameters into a curve after the calculation of 4000 examples. The curve is the Pareto front of double object optimization. All points on the Pareto front could be used as the final optimization result because there are no points with better performances for both than this point in the whole solution group.

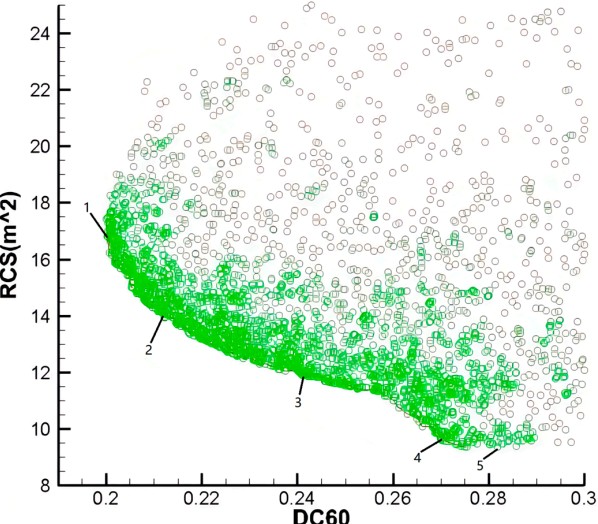

**Figure 13.** Pareto front of multi-objective optimization.

Five sample points marked in Figure 13 were randomly selected from the Pareto front to verify the reliability of the front. Figure 14 shows the Mach number distributions. The RCS values are shown in Figure 15. The backflow zone of each intake was significantly smaller than that of the original intake and the shock wave intensity was remarkably reduced. Besides, five intakes have smaller RCS values than the original intake.

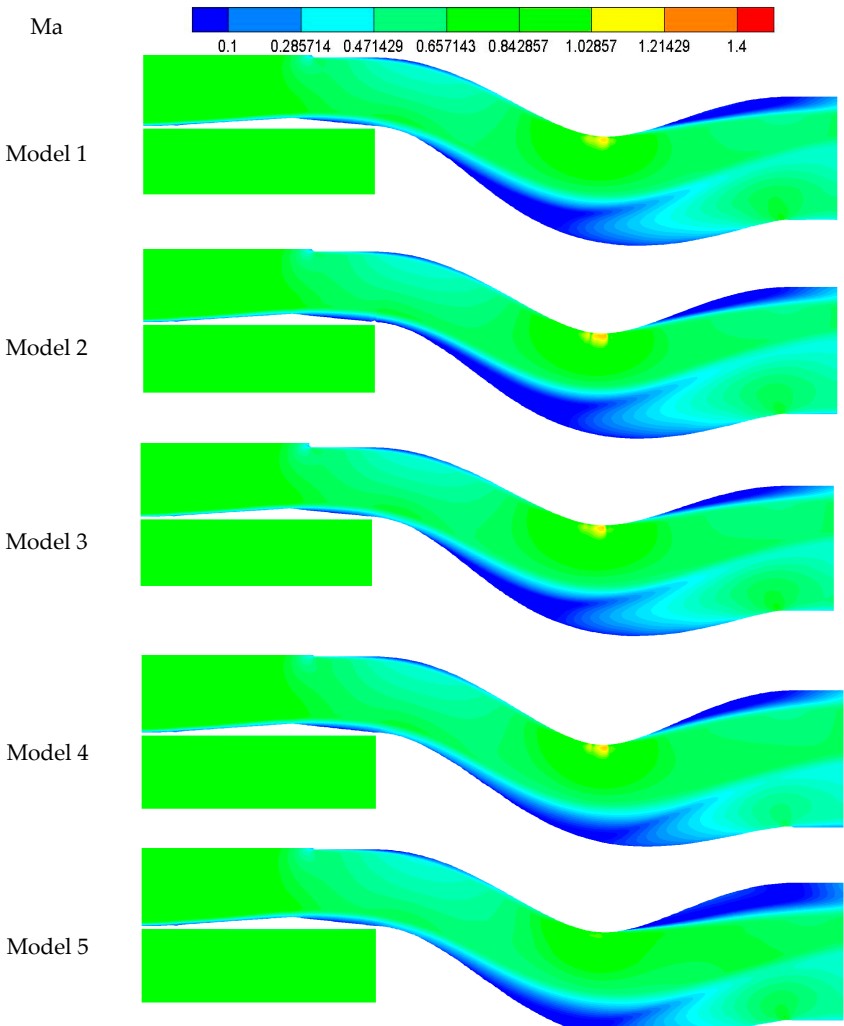

**Figure 14.** Mach number distribution of sample points.

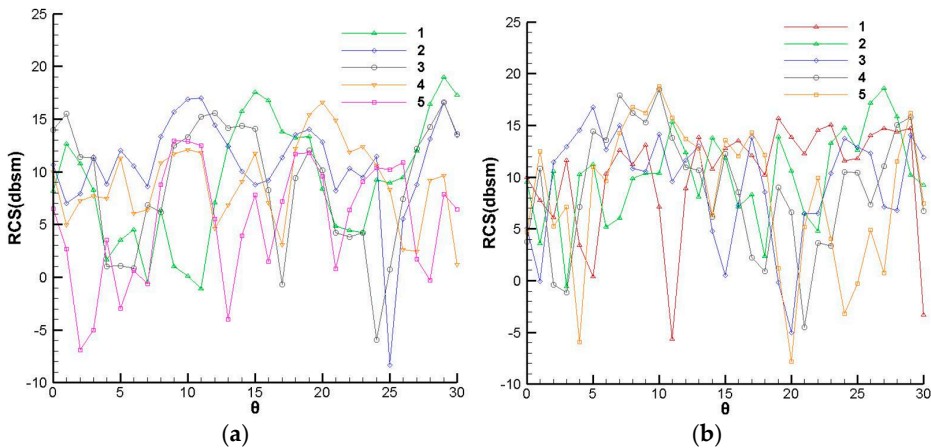

**Figure 15.** RCS values of sample points. (**a**) Horizontal polarization (**b**) Vertical polarization.

The model parameters of sample points are shown in Table 9, while the predicted and actual values of the performance parameters are shown in Table 10. The errors between the calculated values and predicted values of five intakes are generally within an allowable range. The R values of model 4 and model 5 have larger errors, consistent with the results of the error analysis. Therefore, model 1 was selected as the final optimization result, which has both excellent electromagnetic and aerodynamic performance. The swirl distortion index SC60 could be calculated using Equation (7).

$$SC60 = \frac{\overline{V_{max}(60)} - \overline{V_{min}(60)}}{\overline{V^*}} \qquad (7)$$

where $\overline{V_{max}(60)}$ is the minimum secondary flow velocity and $\overline{V^*}$ is the average velocity at the outlet.

**Table 9.** Model parameters of sample points.

| Sample | Y | K | $A_s$ | $A_1$ | $A_2$ |
|--------|--------|--------|---------|---------|--------|
| 1 | 0.1935 | 0.4546 | 1.9230 | 0.1030 | 0.2277 |
| 2 | 0.1797 | 0.6054 | 1.4740 | −0.2675 | 0.1318 |
| 3 | 0.1737 | 0.7093 | 1.3335 | −1.0590 | 0.1974 |
| 4 | 0.1833 | 0.8524 | −0.9254 | −1.7582 | 2.8266 |
| 5 | 0.1687 | 0.8850 | 2.2570 | −0.4626 | 0.4221 |

**Table 10.** Error statistics of performance parameters.

| | DC60 | | M (kg/s) | | R (dbsm) | | TPR | SC60 |
|---|--------|-----------|--------|-----------|--------|-----------|--------|--------|
| | Actual | Predicted | Actual | Predicted | Actual | Predicted | | |
| 1 | 0.2039 | 0.2056 | 37.26 | 37.19 | 12.17 | 11.76 | 0.9511 | 0.0858 |
| 2 | 0.2128 | 0.2156 | 36.98 | 37.65 | 12.16 | 11.34 | 0.9484 | 0.1269 |
| 3 | 0.2516 | 0.2487 | 37.61 | 37.29 | 11.56 | 10.64 | 0.9537 | 0.1247 |
| 4 | 0.2976 | 0.2699 | 37.59 | 37.72 | 11.31 | 9.28 | 0.9531 | 0.0646 |
| 5 | 0.3084 | 0.2855 | 37.16 | 37.96 | 10.44 | 9.17 | 0.9510 | 0.0665 |

### 3.4. Optimization Effect

Three intakes were compared to display the optimization effect, including the straight intake named model A, the original intake named model B, and the optimized intake named model C.

Figure 16 shows the flow field distribution of three intakes. A local low-pressure zone appeared below the outlet of model A and above the outlet of model B, but was non-existent at the outlet of model C, indicating that DC60 was reduced effectively. Besides, the local shock wave obviously appeared at model B and was inconspicuous at model C, indicating that the total pressure loss was also reduced. Figure 17 shows the RCS values in the vertical direction. At both polarization modes, the maximum RCS value of model A exceeded 20 dB, while the RCS values of most angles were larger than those of model B and model C, indicating that the electromagnetic performance was improved by connecting the S-duct diffuser. In addition, the maximum RCS value of model C is 3 dB lower than that of model B with vertical polarization but only 1 dB higher than that of model B with horizontal polarization.

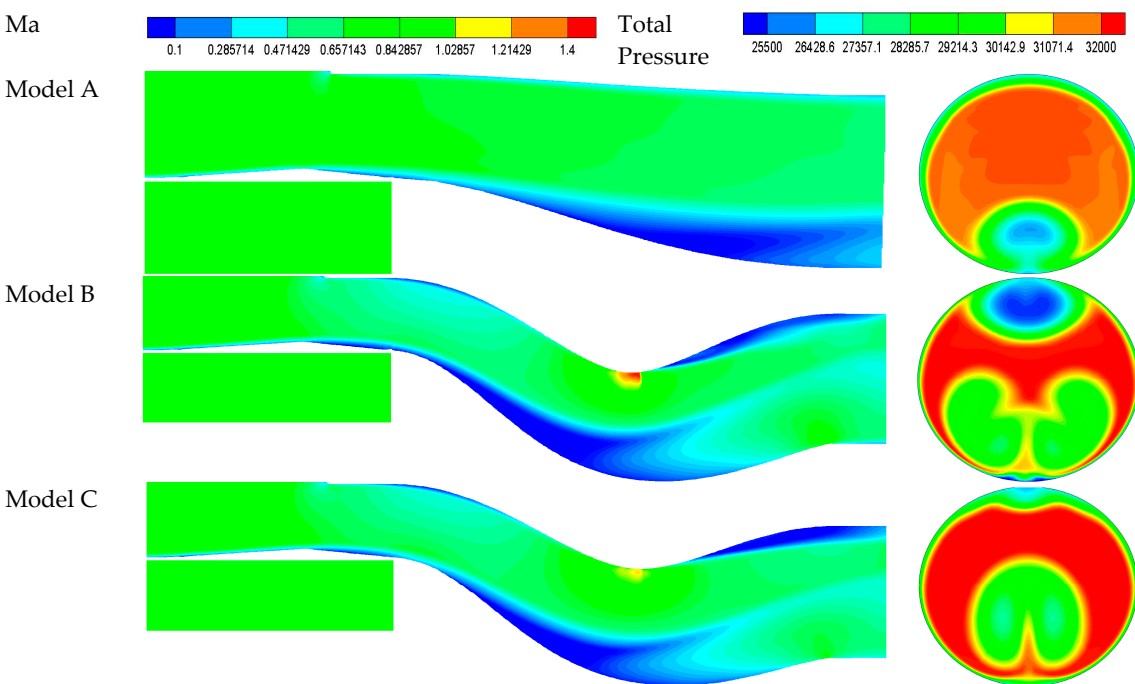

**Figure 16.** Distribution of flow field.

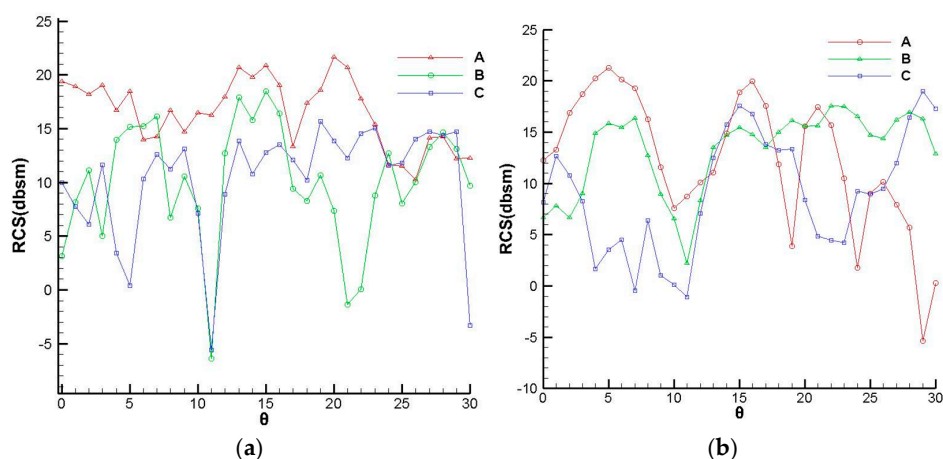

**Figure 17.** RCS in the vertical direction. (**a**) Horizontal polarization (**b**) Vertical polarization.

Six performance parameters of the three intakes are summarized in Table 11. By comparing the values of model A and model B, it is indicated that the loss of aerodynamic performance is usually the cost of improvement in electromagnetic performance with the un-optimized S-duct diffuser. In addition, compared with the values of model B, the $R_a$ value of model C decreased by 2.39, the SC60 value increased by 0.09, the M value increased by 2.6 kg/s, and the DC60 value decreased by 0.24, indicating that aerodynamic performance and electromagnetic performance of S-duct intake were improved after optimizing.

**Table 11.** Result statistics.

| Model | $R_a$ (dbsm) | $R_b$ (dbsm) | DC60 | M (kg/s) | TPR | SC60 |
|:---:|:---:|:---:|:---:|:---:|:---:|:---:|
| A | 17.523 | 15.637 | 0.3810 | 37.11 | 0.9498 | 0.0394 |
| B | 14.45 | 12.64 | 0.4410 | 34.64 | 0.9177 | 0.1739 |
| C | 12.06 | 12.27 | 0.2039 | 37.26 | 0.9511 | 0.0858 |

### 3.5. Verification at Off-Design Conditions

The flow field of model C was calculated at different Mach numbers of 0.6–1.1, with intervals of 0.1. The static pressure of the outlet was changed to 20,000 Pa, while the other boundary conditions remained unchanged. Figure 18 shows the total pressure distributions of the outlet at six conditions. At each condition, two low-pressure zones appeared at the top and bottom of the outlet, respectively, and the areas of low-pressure zones increase with the increase of the Mach number, which conforms to the theoretical expectation. Three aerodynamic parameters at six off-design conditions are shown in Figure 19, including DC60, flow coefficient ($\psi$), and TPR coefficient ($\sigma$). The flow coefficient refers to the ratio of the actual flow to the ideal design flow. The DC60 increases firstly and then decreases with the increase of the Mach number, which is greater than 0.4 when the Mach number exceeds 0.7. Besides, the TPR coefficient decreases with the increase of the Mach number. Nevertheless, the flow coefficient and TPR coefficient are always greater than 0.9 at all six conditions. In summary, the intake is applicable for conditions with Mach numbers from 0.6 to 1.1.

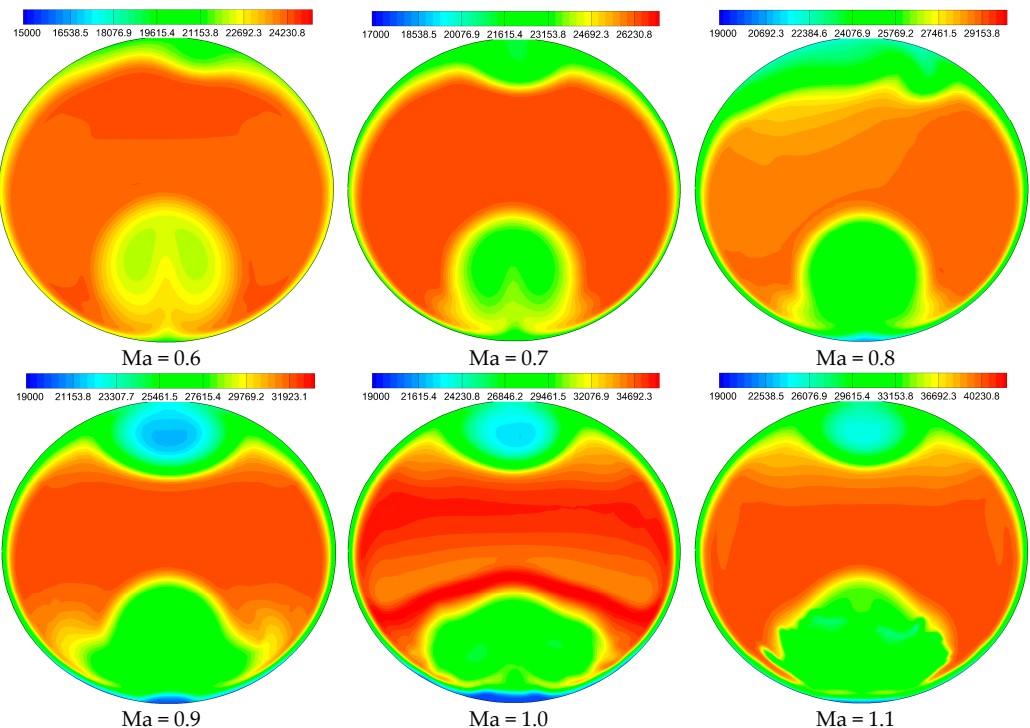

**Figure 18.** Total pressure of outlet at off-design conditions.

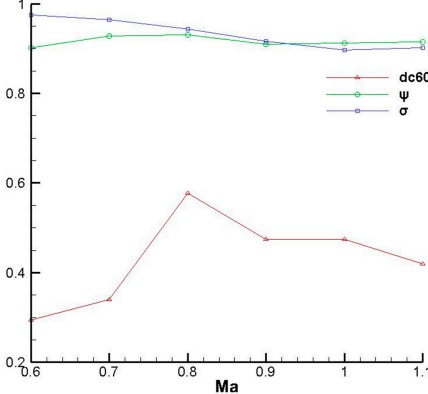

**Figure 19.** Aerodynamic performance at off-design conditions.

The electromagnetic echo intensity of the cavity can be effectively reduced by coating the surface with low scattering material [21]. A material with a relative magnetic permeability $\mu = 1.29 - 0.57j$ and relative dielectric constant $\varepsilon_r = 9.72 - 1.08j$ was coated on all surfaces of the model except on the outlet section. The frequencies were set as 3 GHz and 10 GHz.

Figure 20 shows the RCS values at various calculation conditions, and the statistics are counted in Table 12. At the four conditions with a frequency of 3 GHz, the maximum RCS values were all less than 10 dB and the RCS values of most detection angles were at a low level, indicating that the S-duct intake coated with low scattering material has excellent electromagnetic performance. At the condition with a frequency of 10 GHz and vertical polarization, the RCS value with detection angle of 0° is inadequately greater than 15 dB. Therefore, some necessary measures should be taken to further improve the electromagnetic performance at this condition. At other conditions, the same conclusion could be obtained. According to the table, the average RCS values were all less than 3 dB at various conditions.

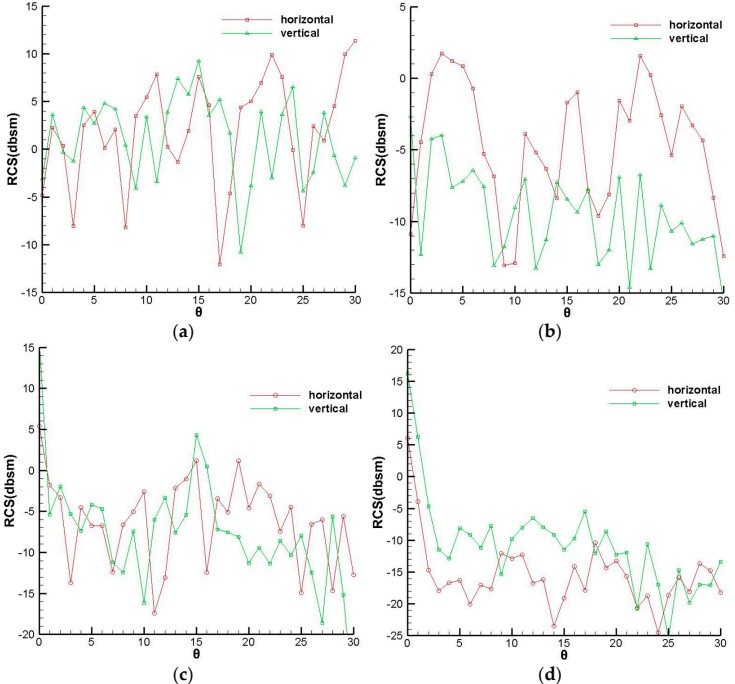

**Figure 20.** RCS of inlet coated with low scattering material. (**a**) Vertical detection-3GHz (**b**) Horizontal detection-3GHz (**c**) Vertical detection-10GHz (**d**) Horizontal detection-10GHz.

**Table 12.** Statistics of RCS (dbsm) at various conditions.

| Frequency | 3 GHz | 10 GHz |
|:---:|:---:|:---:|
| V-V | 4.70 | −3.33 |
| V-H | −2.82 | −7.75 |
| H-V | 2.99 | 2.17 |
| H-H | −8.39 | 1.97 |

## 4. Conclusions

1.  The S-duct diffuser can effectively reduce the electromagnetic echo intensity of the intake but generally results in the loss of aerodynamic performance. The backflow, the second flow, and the local shock wave occurred in the diffuser due to the unsuitable design, decreasing the mass flow rate and the total pressure loss while increasing the total pressure and the swirl distortion;

2. By determining the five parameters with the quartic polynomial functions, a definite S-duct intake model could be constructed, causing the optimal design to be feasible. According to the samples selected with the Optimized Latin Hypercube method, four effective models based on the Kringing model were established, with satisfactory accuracy for the aerodynamic performance parameters and available accuracy for RCS. By using the global multi-objective optimization algorithm, five solutions were obtained and verified to be feasible, with one of them set as the optimal result;

3. By the optimal design stage, the two performances of the S-duct intake were both improved, with the area of the low-speed zone decreasing, the intensity of the local shock wave reduced, the second flow suppressed, and the average RCS decreased by 2 db. Compared with the straight intake, the electromagnetic performance of the optimized S-duct intake was obviously improved and the aerodynamic performance was similar except that the SC60 value was increased by 0.05. After verification, the optimized intake also has both applicable aerodynamic and electromagnetic performances at various off-design conditions.

**Author Contributions:** Conceptualization, B.W.; methodology, B.W.; validation, B.W.; formal analysis, B.W.; investigation, B.W.; resources, B.W.; data curation, B.W.; writing—original draft preparation, B.W.; writing—review and editing, Q.W.; visualization, B.W.; supervision, Q.W.; project administration, Q.W. All authors have read and agreed to the published version of the manuscript.

**Funding:** This research received no external funding.

**Institutional Review Board Statement:** Not applicable.

**Informed Consent Statement:** Not applicable.

**Data Availability Statement:** Not applicable.

**Conflicts of Interest:** The authors declare no conflict of interest.

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
