# Peer review of "Numerical Optimization of Electromagnetic Performance and Aerodynamic Performance for Subsonic S-Duct Intake"

_aerospace, doi:10.3390/aerospace9110665_

Round 1
Reviewer 1 Report
The introduction provide background and includes relevant references, but which need to be supplemented with the results of each mentioned research.
Review terminology such as "recently", when often referring to old references.
Better justify the input data chosen to perform the simulations and the proposed algorithm.
Do not rely solely on a reference to determine whether a parameter A or B is the most important and recommended to perform the simulations.
Review the quality of multiple figures (tagged in PDF)
Review the conclusions, since compiling only the results obtained in the simulations does not mean a critical evaluation of the results. For example, how could the authors interpret these results to propose a truly favorable operating condition?
The algorithm can obtain a variety of solutions, but are they all feasible? Ideally, you should evaluate your use based on more evidence to validate the results of use.
Overall, the paper has merit and is interesting to the magazine's audience, requiring only minor adjustments.

Author Response
Dear Reviewers:
We are very grateful to reviewer for reviewing the paper so carefully.We have tried our best to improve and made some changes in the manuscript.Please see the attachment.
Best Regards!
PhD student Bin Wang and Prof. Qiang Wang

Reviewer 2 Report
I am not a radar cross section expert, so I am not in a good position to evaluate the validity of the RCS analysis in this paper. However, conceptually, this multiobjective study is well-posed. Not much is written on RCS in the multidisciplinary design optimization literature so this is a welcome contribution.
The fit of the Kriging models is not very good. It would be better to either use a Kriging model which can update as optimized points are found (ie, added to the training set), or to use a more refined hypercube experiment (plus potentially adjusting kriging hyperparameters or using another kriging variant to improve goodness of fit). Because of the relatively poor kriging fit I suspect that the optimizer is missing some better designs. The study would have been improved with more geometric degrees of freedom. But the paper is mostly acceptable as is.
Author Response
Dear Reviewers:
We are very grateful to reviewer for reviewing the paper so carefully. Thanks for your suggestions! As you said, the Kriging model does not fit the RCS very well. We are trying your method and more mathematical models. If some achievement are made, it will be shown in the new paper.
Best Regards!
PhD student Bin Wang and Prof. Qiang Wang
Reviewer 3 Report
I strongly recommend considering the following aspects to improve the paper:
- General quality of figures: Pls. replace figures 2, 3, 6, 12, 14, 16 which such of better quality. Avoid changing the aspect ration of the figures compared tho their origin.
- P6 "Therefore, the numer of Mesh used to calculate was set to 2million". Was does that mean, in table 4 there is a Mash A, B and C - I guess mesh-B has beeen selected?
- I do not uderstand why figure 7 shall show that Mesh-A result are "significantly different" form B and C - pls. clarify
- the optimization (table 8) does obviouslay lead to changes in the inlet mass flow (31,4 up to 37,7 kg/s). Why is this a open parameter? From my understanding it should be a constraints with very small range (if any) since it affects the engine performance significantly.
- is the total pressure loss identical for all designs shown in table 8 ? my expectation is that is does change - if that is th case, pls. shown it in table 8
- DC60: Pls. give the definition of the DC60 you are using. Pls. comment on what DC60 levels are expected or maybe comtraint for the engine since 0.24 to 0.57 is all very high using the typical definiton. I woul appreciate discussing the DC60 level in the light of other publications (same with R) rather than just showing the relative improvement from the optimization
- figure 13 and 14: pls. indicate the 5 candidates shown in figure 14 also in figure 13 (maybe that has been done but due to the poor quality it cannot be identified)
- p13: I assume that Model C has been chosen for the Off-design analysis in hapter 3.5. However, this is not fully clear.
- p.12/13: Comparing the flow filed o Model B to Model C it appears that for B the DC60 is dominated by the 12 o#clock position with its low pressure area while for Model C the DC60 is driven by the low part of the flow field. Here the typical "owl face" can be identified but doesn't that mean that also swirl plays a significant role for this pattern ? Pls. comment on why swirl or descriptors like SC60, SCI are not used for this analysis.
Author Response

(The authors gave the same response as above.)
